# Using Machine Learning for Remote Behaviour Classification—Verifying Acceleration Data to Infer Feeding Events in Free-Ranging Cheetahs

**DOI:** 10.3390/s21165426

**Published:** 2021-08-11

**Authors:** Lisa Giese, Jörg Melzheimer, Dirk Bockmühl, Bernd Wasiolka, Wanja Rast, Anne Berger, Bettina Wachter

**Affiliations:** Leibniz-Institute for Zoo- and Wildlife Research, Alfred-Kowalke-Straße 17, 10315 Berlin, Germany; lisagiese89@gmail.com (L.G.); melzheimer@izw-berlin.de (J.M.); dirkbockmuhl@yahoo.com (D.B.); officebernd@gmail.com (B.W.); rast@izw-berlin.de (W.R.); wachter@izw-berlin.de (B.W.)

**Keywords:** accelerometry, automated behaviour classification, *Acinonyx jubatus*, cheetah, GPS clusters, supervised machine learning

## Abstract

Behavioural studies of elusive wildlife species are challenging but important when they are threatened and involved in human-wildlife conflicts. Accelerometers (ACCs) and supervised machine learning algorithms (MLAs) are valuable tools to remotely determine behaviours. Here we used five captive cheetahs in Namibia to test the applicability of ACC data in identifying six behaviours by using six MLAs on data we ground-truthed by direct observations. We included two ensemble learning approaches and a probability threshold to improve prediction accuracy. We used the model to then identify the behaviours in four free-ranging cheetah males. Feeding behaviours identified by the model and matched with corresponding GPS clusters were verified with previously identified kill sites in the field. The MLAs and the two ensemble learning approaches in the captive cheetahs achieved precision (recall) ranging from 80.1% to 100.0% (87.3% to 99.2%) for resting, walking and trotting/running behaviour, from 74.4% to 81.6% (54.8% and 82.4%) for feeding behaviour and from 0.0% to 97.1% (0.0% and 56.2%) for drinking and grooming behaviour. The model application to the ACC data of the free-ranging cheetahs successfully identified all nine kill sites and 17 of the 18 feeding events of the two brother groups. We demonstrated that our behavioural model reliably detects feeding events of free-ranging cheetahs. This has useful applications for the determination of cheetah kill sites and helping to mitigate human-cheetah conflicts.

## 1. Introduction

One of the most complex aspects to study in animals is their behaviour [1]. Understanding species-specific behaviour is essential for their management, welfare and conservation [2,3,4,5,6]. Studying the behaviour of free-ranging animals often poses several challenges. Traditional ethology is based on direct observation, but the presence of an observer might alter the natural behaviour of the study animals if they are not fully habituated to the observer [7]. Furthermore, the animals might be difficult to observe when they are nocturnal, shy and cryptic [8], or are from an aquatic or migratory species [9]. 

Using biologgers has fueled research on remote monitoring of animal behaviour and unlocked the former limitations of direct observations [10,11,12]. Collecting data on fine-scale movement via accelerometers (ACCs) is an effective way to continuously monitor free-ranging animals [13,14]. ACCs measure the change in velocity of the sensors attached to the body and allow translation of the quantified movement patterns into distinct behavioural categories or activities [13]. This technique has been successfully used for a variety of bird species [15,16,17,18,19], marine animals [20,21,22,23], and terrestrial mammals [24,25,26,27,28], including cheetahs, *Acinonyx jubatus* [29,30,31,32]. 

The training of machine learning algorithms for pattern recognition and data classification is conducted with observational verified data. For this, a ground-truthed behavioural data set is used to train the algorithm and another data set is used to infer behaviour from it. Campbell and colleagues [30] used ground-truthed data of a tame surrogate dog (*Canis lupus familiaris*) to train their support vector machine (SVM) algorithm to predict walking, running, sitting, standing and lying in ACC data collected from different species, including the cheetah. Using the cheetah dataset generated by Campbell et al. [30], Bidder and colleagues [31] successfully predicted sitting and standing in cheetahs using another algorithm, the k-nearest neighbour (KNN) algorithm. While these two studies had strict validation purposes, Grünewälder et al. [29] observed six free-ranging cheetahs and recorded stationary, mobile and feeding behaviour during an average time of 31 h. A SVM and a hidden Markov model were trained with this ground-truthed dataset of each cheetah and used to predict the three behaviour categories for the remaining unobserved data collected during an average time of 332 days of the six cheetahs [29]. To our knowledge, there is currently no study that transferred a behaviour classification model trained on one group of cheetahs to another group of cheetahs. If the inferred behaviour classification in the latter group is verified, the classification model has a wide application for the species.

Cheetahs occur at low densities and cover large home ranges, making it difficult to observe them in the wild [33,34,35]. Southern Africa hosts the largest free-ranging cheetah population in the world, where cheetahs predominantly occur on farmland, i.e., outside of protected areas such as national parks or game reserves [36,37,38]. On farmland, cheetahs might come into conflict with landowners when they predate on their livestock and/or valuable game species [35,39]. Because cheetahs are not protected on farmland, landowners regularly kill cheetahs to prevent such losses [35,39]. The cheetah is listed as vulnerable by the International Union for Conservation of Nature and Natural Resources (IUCN) [36], and human–cheetah conflicts are amongst the major threats to its existence in the wild.

To investigate the extent of the farmer–cheetah conflict, it is important to understand the feeding ecology of the cheetahs, thus detecting feeding behaviours in the wild. Cheetahs feed very rarely on carcasses, thus feeding events can be interpreted as cheetahs having killed the prey animals [40]. Grünewälder et al. [29] used individual-based behaviour predictions from ACC data to investigate feeding-related behaviours. Other studies successfully used clusters of GPS positions to detect potential kill sites of carnivores, e.g., of leopards (*Panthera pardus*) [41,42] and jaguars (*Panthera onca*) [43]. Such clusters represent locations to which the animal returned repeatedly and/or spent an extended amount of time. From these spatial data alone, it is not possible to determine whether the animal was indeed feeding or rather resting. It is therefore required that GPS positions are retrieved and visualised in regular intervals and detected clusters visited within a time frame that allows finding of prey remains in the field [41,42]. Additional spatial and temporal information can be helpful in discriminating between potential kill sites and non-kill sites [43]. Combining both ACC and GPS data, Wang et al. [44] were able to identify and verify five predation events in free-ranging cougars (*Puma concolor*) by associating periods of high acceleration movements with subsequent GPS clusters. This was a useful approach because their behavioural model was trained with ground-truthed data of captive pumas and was weak at predicting feeding behaviour. Also, the combination of ACC and GPS data was successfully used to differentiate feeding from non-feeding events in a single leopard [45] and non-active (resting) from active (potentially feeding) stationary events in cougars [46].

The goal of this study was to detect six behaviours, including feeding behaviour, from acceleration data recorded in free-ranging cheetahs by using behaviour classification models trained on ACC data of captive cheetahs. For this, we first tested the potential of six common supervised machine learning algorithms (MLA) in inferring the behaviours from ACC data from captive cheetahs. We trained the MLAs on a dataset which we ground-truthed by direct observation. We then used the validated dataset to identify behaviour in ACC data from free-ranging cheetahs and, by adding GPS cluster analysis, detect feeding events. We verified feeding events identified by the model with kill sites that were previously confirmed by visiting GPS clusters in the field and identifying the prey remains.

## 2. Materials and Methods

### 2.1. Study Area and Animals

For the data validation, we used five captive cheetahs held in two enclosures on a private game farm in east-central Namibia (−22.5803° S, 18.1875° E). The enclosures were set within their natural environment. One enclosure (1.0 ha) contained three brothers of approximately 8 years of age and the other enclosure (1.3 ha) a brother and sister of approximately 4 years of age. All five animals were born in the wild but came into captivity at an early age. They were accustomed to human presence and allowed observers in their vicinity (Figure 1). The animals had *ad libitum* access to clean water in an artificial water trough and were fed once a day in the late afternoon. On 3 October 2017, we fitted GPS collars with ACC sensors (e-obs GmbH, Grünwald, Germany [47]) on the three brothers during immobilisations for medical check-ups and on the two other cheetahs without immobilisation since they accepted this handling by one of the authors (DB). The collars were colour-coded to help with the identification of the animals during the behavioural observations. After data collection, the collars were removed again with the same procedures.

To test the applicability of the behavioural model, we used data from four free-ranging territorial males in two coalition groups. They were equipped with the same collar types as the five captive cheetahs on 18 November 2014 and 7 December 2014 (coalition 1) and 21 May 2014 (coalition 2). Their GPS data were used to visually detect kill sites (by GPS clusters) and verify them by checking for prey remains in the field. Their ACC and GPS data were then used to detect feeding events correlating to the confirmed kill sites.

All experimental procedures described were approved by the Internal Ethics Committee of the Leibniz Institute for Zoo and Wildlife Research (IZW, Berlin, Germany) (permit number: 1 April 2002) and the Ministry of Environment, Forestry and Tourism of Namibia (permit numbers: 1514/2011, 1689/2012, 1813/2013), 1914/2014, 2067/2015, 2194/2016, 2208/2017).

### 2.2. Data Collection

#### 2.2.1. Data Collection on Captive Cheetahs for Model Validation

Observations of the three brothers were conducted during 36 full or half days between the 6 October and the 10 December 2017, whereas the brother and sister were observed during 20 full or half days between the 20 November and the 19 December 2017. The e-obs collars were set to record ACC data from 4:00 until 17:00 Universal Time Coordinated (UTC), corresponding to the hours of daylight at the time of data collection to ensure good visibility for the behavioural observations. The ACC data were recorded in ‘bursts’ occurring every 30 s. The collars were equipped with two different tag firmware and the settings for the burst lengths and sampling frequencies were chosen from the configuration options provided by the manufacturer. We programmed the collars in such a way that the dataset collected with the two firmwares were as similar as possible and fitted best with the data collected from the free-ranging animals (Table 1). The sampling frequency for the three captive brothers was temporarily increased from 10 Hz to 33.3 Hz during 7 to 15 days to assess differences in model performance due to different sampling frequencies in another study. To also make use of the 33.3 Hz data in the present study, we resampled the data using the resample_poly function from the signal package within the scipy library (Version 1.4.1 [48]). The resampling was done in python 3.7.10 [49]. This function includes a low pass filter that is applied before down sampling. The collars did not have an analog anti-aliasing filter, but these low sampling frequencies were nevertheless chosen (see Section 4). We focused on the heave-axis (z-axis, Figure 1) because the collars of the free-ranging cheetahs only recorded ACC data from this axis (see Section 2.2.2). 

All observations were carried out by two observers simultaneously. An ultra-high frequency (UHF) pinger with a unique frequency for each collars was running during the on-time period, emitting a rhythmical beeping signal which was picked up by a UHF receiver (AR8200, AOR, Tokyo, Japan). A change in the rhythm of the beeping signal indicated the start and the end of each ACC burst. During the burst, the behaviours of the animals were observed and recorded on a sheet. Using a digital radio clock, each recorded burst was linked to a unique timestamp. Since the collars of the cheetahs sharing the same enclosure were synchronised in their sampling time of ACC data, we recorded the behaviour of all individuals in sight during any given burst.

We recorded six behavioural categories during the observations: drinking (D), feeding (F), grooming (G), resting (R, including all non-motion behaviours), trotting/running (T) and walking (W). To simulate feeding events as naturally as possible, we used either big pieces of meat, bones and organs or entire carcasses obtained from animals hunted on the game farm. We aimed to record at least 30 bursts per behaviour and animal to ensure good trainability of the MLAs. The ACC data were downloaded manually from the collars with a UHF connection using an e-obs basestation (BaseStation b5) [47] at the end of each observation day.

For each behaviour, one ACC burst was randomly chosen to create sample plots of the raw data (Appendix A Figure A1).

#### 2.2.2. Data Collection on Free-Ranging Cheetahs to Test Model Application

For the model application part, GPS and ACC data from four collared free-ranging cheetah males in two coalitions were used over 91 days from 6 February 2015 to 7 May 2015. The collars recorded ACC data of the heave-axis (z-axis, Figure 1) in bursts of 3.6 s every two minutes at a frequency of 10 Hz (Table 1). Due to battery lifetime constraints and fast, field-applicable data download, we focused on one axis. We decided on the heave-axis, because our main interest was to detect feeding behaviour and the ripping movements of the head during feeding are likely to be well display in this axis. In each coalition, GPS data were recorded every 15 min from both coalition members, while for one coalition member the schedule was increased temporarily to every three minutes. The GPS collars with the higher resolution, recorded fixes from the 22 January 2015 to the 16 April 2015 for coalition 1 and from the 6 February 2015 to the 25 March 2015 for coalition 2, respectively. This higher frequency of GPS data recording was used to facilitate a higher accuracy in determining a feeding event and the duration thereof. To save battery lifetime we used an ACC informed GPS recording scheme, such that GPS data were recorded only every 6 h when a predetermined threshold of ACC data variance between consecutive GPS fixes were not reached, implying the animal was inactive. As soon as the animal became active again, the GPS interval raised again to 15 min or 3 min, respectively. Due to the low battery lifetime of one member of coalition 2, we set the GPS interval from 3 min to 6 h for the last 12 days. 

The GPS data were downloaded on a weekly basis during aerial tracking flights as described in Melzheimer et al. [34]. When initially analysing the GPS positions on a map using geographical information systems, distinct clusters were identified visually at places where the cheetahs spent an extended amount of time at the same geographical location. Due to the ACC informed GPS measurement, the collars only recorded consecutive GPS fixes at the same position if the animal was active. The dynamic GPS scheduling was therefore used as an additional clue to distinguish resting and feeding spots. Once such potential feeding spots were identified, the sites were visited within a week of the suspected kill and examined for prey remains to confirm that a prey animal was killed and eaten there. The duration of the feeding event was set based on the first and last GPS position taken at the specific location where the prey was consumed. In cheetahs, members of the same coalition are always together [34]. Thus, we assigned feeding events of members of the same coalition to a common kill site. 

### 2.3. Data Analyses

#### 2.3.1. Algorithms Used and Data Validation

We worked with supervised machine learning algorithms (MLAs) to analyse the ACC data [50]. Supervised MLAs are common tools in pattern recognition and are characterised by the two processes of training and testing. In short, we used six supervised MLAs, namely linear discriminant analysis (LDA) [51], quadratic discriminant analysis (QDA) [16], the KNN algorithm [31,52], the classification and regression tree (CART) [16,53] algorithm, SVM algorithm [54,55], and the random forest (RF) [56,57] algorithm. All six MLAs were applied on the same set of six predictors: mean of z-axis (mnz), standard deviation of z-axis (sdz), weighted mean of the Fourier transformation of the z-axis (wmz = weighted mean of the periodogram ordinates for all Fourier frequencies w = 2πj/n, j = 1,..., q with q = n/2 (n even) or q = n/2 − 1 (n odd), with n being the number of data points in the sample), inverse coefficient of variance of z-axis (ICVz = mnz / sdz), kurtosis of z-axis (kz) and skewness of z-axis (sz). They were calculated for each burst of raw ACC data. We also implemented two ensemble learning approaches, the mean and the majority voting, that are based on the predictions derived from the six algorithms to improve the classification performance of our behavioural model. More details are provided in the Appendix A Text A1.

The validation of ACC data was done using two different approaches. Firstly, we estimated the classification performance of our behavioural model with a classic leave-one-out cross-validation (LOOCV) using the complete dataset of the five captive cheetahs, i.e., the ground-truthed dataset. Secondly, we conducted a per-animal cross-validation (PACV) for each of the five cheetahs by training the behavioural model with the complete datasets of four animals to predict behaviour in the complete dataset of the fifth animal.

To further improve the performance of the model, we added a probability threshold. For this the model labelled a burst as a particular behaviour only when an algorithm classified this behaviour with the defined or a higher probability. Any burst below the probability threshold was labelled “not conclusive”. To determine the probability threshold with the highest gain in predictive performance in relation to the percentage of bursts that did not exceed the probability threshold, we performed the LOOCV with thresholds ranging from 0.3 to 0.8. For the PACV we only used the probability threshold that performed best. For the LOOCV, we assessed the performance of each algorithm and of the two ensemble approaches by calculating overall precision, precision per behaviour with PR = TP/(TP+FP) and recall per behaviour using the respective confusion matrices, with RE = TP/(TP+FN) (TP = true positive, FP = false positive, FN = false negative). For the PACV, we did the same but assessed the performance only for the two ensemble approaches since they performed best in the LOOCV (Table 2). 

#### 2.3.2. Model Application and Verification 

To test the applicability of the behavioural model, we used ACC and GPS data of the four free-ranging cheetahs, covering one full day before and one full day after the date of each confirmed kill site (see Section 2.2.2) (that were previously identified by visually checking for GPS clusters and visiting these clusters in the field to check for prey remains). This time-frame was based on the average kill rates of free-ranging cheetahs, which is less than, or equal to, one kill every two days in southern Africa [29,58] and never exceeded a kill per day in the Serengeti National Park in Tanzania, East Africa [33]. Using this time frame, we limited the chances of including data from unconfirmed and unverifiable feeding events.

After calculating the six predictors for all raw ACC bursts of the free-ranging cheetahs, we trained the behavioural model with the previously established complete ground-truthed dataset to predict behaviour in this unknown data using the six algorithms, the two ensemble learning approaches and the probability threshold that performed best. The resulting file contained all eight behaviour predictions (from the six MLAs and the two ensemble learning approaches) for each burst. The unique timestamp of each burst allowed us to ‘read’ the predicted behaviours in chronological order.

Next, we searched for clusters of feeding bursts in the chronological sequence of all predicted behaviours, in the following termed as ‘feeding cluster’. Since both the ensemble learning approaches with probability threshold and the SVM algorithm with probability threshold performed best in predicting feeding behaviour in the LOOCV (Table 2), all bursts that were identified to be ‘feeding’ by either one of the ensemble learning approaches or the SVM algorithm were defined to be ‘true feeding’. We used a sliding window approach to find feeding cluster with at least a third of bursts per sliding window being ‘true feeding’. We determined start and end time of these feeding clusters allowing a minimum length of 30 min per cluster to cover medium to large kills following Grünewälder et al. [29]. They determined for cheetahs a feeding time of 10 to 15 min for small prey animals and one of up to 3 h for large prey animals. We allowed for a maximum gap of 120 min of ‘not feeding’ within a cluster. 

We then screened the corresponding GPS data for clusters. GPS clusters were determined based on the condition that consecutive coordinates did not exceed a distance of 50 m to each other and lasted at least 30 min. The centroid of each GPS cluster was determined by calculating the mean longitude and latitude of all coordinates of the cluster. 

The GPS clusters and feeding clusters were then checked for overlaps in time. If multiple GPS clusters fell into the time frame of one feeding cluster, we either merged these GPS clusters, if their centroids were less than 50 m apart from each other, or we determined the GPS cluster with the highest time overlap with the feeding cluster to match a feeding event. If this process was inconclusive, we examined the data manually and assigned GPS coordinates to the ‘unmatched’ feeding cluster or dropped the feeding cluster as ‘false positive’. If feeding clusters started/ended more than 3 or 15 min, respectively (i.e., the interval between consecutive GPS fixes) before/after the animal was stationary according to a GPS cluster, start and end time of the feeding cluster were corrected accordingly. 

Since free-ranging cheetahs kill their prey themselves and only rarely scavenge [40], they typically engage in hunting activity prior to a feeding event. We therefore also scanned the behaviour predictions one hour prior to the feeding events for bursts identified as ‘high ACC variation’, i.e., trotting and running, indicating a potential hunt. 

All feeding events determined this way were then descriptively compared to the previously confirmed nine kill sites.

All analyses were conducted using the statistical computing software R Version 3.5.1 [59].

## 3. Results

### 3.1. Data Validation

For the five captive cheetahs, a total of 7760 bursts were recorded, of which 6,641 were single-behaviour bursts. The remaining 1119 bursts were discarded as they contained mixed behaviours, i.e., a change from one behaviour to another within a burst. The most sampled behaviour was resting with 2717 bursts (35.0%), followed by walking with 1673 bursts (21.6%), feeding with 1319 bursts (16.9%), and grooming with 487 bursts (6.3%). Trotting/running and drinking were the least sampled behaviours with 260 (3.4%) and 185 (2.4%) bursts, respectively. 

The LOOCV was run with thresholds from 0.3 to 0.8 to find the most effective one. The probability threshold of 0.5 had the best ratio in increasing the precision and at the same time decreasing the recall (Appendix A Table A1). Confusion matrices for all algorithms with the 0.5 probability threshold are in the appendix (Appendix A Table A2, Table A3, Table A4, Table A5, Table A6, Table A7, Table A8 and Table A9). Adding this threshold to the LOOCV increased the performance of the model, particularly in predicting feeding behaviour by the SVM algorithm (performance increase 2.7%) by simultaneously only missing 5.4% of true feeding bursts by labeling them as “not conclusive” (Appendix A Table A1). Thus, the SVM algorithm played an essential role in the determination of ‘true feeding’ in the model application. 

With the threshold of 0.5, the LOCCV achieved overall precisions ranging from 57.5% to 91.4%, depending on the algorithm and ensemble learning method used (Table 2). The algorithms achieved high accuracies for resting, walking and trotting/running behaviour, but performed less well for drinking and grooming behaviour (Table 2). The combination of precision and recall for feeding behaviour was best at (78.8%/82.4% (precision/recall)) with the SVM algorithm (Table 2). The two ensemble learning approaches, both performed better than the SVM in respect to the precision but worst in respect to the recall (Table 2). The highest precisions for behaviours were 99.0% and 100% for resting and trotting/running at a recall of 90.7% and 99.2%, respectively. A comparison of all values in Table 2 with the results for the LOOCV without adding a probability threshold is presented in Appendix A Table A10. 

The PACV with a probability threshold of 0.5 also reached high overall precision, ranging from 69.9% to 92.5% for the two ensemble learning approaches, while the overall recall, ranging between 57.5% and 70.0%, did not reach such high values (Appendix A Table A11). The precision for feeding behaviour ranged from 65.6% to 91.4% and the recall from 47.0% to 81.1%.

### 3.2. Model Application

The visual inspection of the GPS data of the two free-ranging male coalitions for clusters and the checking of these spots in the field resulted in nine GPS clusters that we confirmed to be kill sites of the cheetah males because we found in all cases prey remains at the sites. 

We successfully identified in the unknown ACC data set of the two coalitions all nine kill sites and 17 of the 18 feeding events (Table 3). The time ranges of these feeding events fitted into the timespans of the feeding events deduced from the confirmed kill sites. Differences in the durations of GPS clusters between the feeding events detected visually and the model application was based on the different approaches of determining the GPS clusters, i.e., visual vs. automated approach (Table 3). We identified bursts with high ACC variation in the hour before the respective feeding event in eight out of 17 cases (47.1%) and in seven out of nine kill sites (77.8%, Table 3). We also identified one feeding event that was not confirmed in the field with prey remains, although a GPS cluster was identified (Table 3, Figure 2).

Figure 2 illustrates nine days of GPS coordinates of two males of coalition 2 with GPS intervals of 3 min and 15 min, respectively. Both males revealed almost an identical movement path, and the same feeding and resting clusters were detected in the corresponding ACC data. While two feeding events (marked red in Figure 2) were previously confirmed in the field with prey remains, the third one (marked orange in Figure 2) was not confirmed in the field, but was identified in both males. Walking events were only detected in the male with the 3-min GPS schedule. All walking clusters corresponded with lacking stationarity, i.e., no GPS clusters, indicating that the animal was indeed on the move. The calculation of resting and walking clusters is described in Appendix A Text A2.

## 4. Discussion

In this study, we validated a method to remotely and automatically detect with MLAs feeding events/kill sites of free-ranging cheetahs by using ACC and GPS data (Figure 3). We used a two-step process by demonstrating in a first step the applicability of ACC data to predict basic behaviours in five captive cheetahs by using six supervised MLAs. To improve prediction accuracy, we introduced two ensemble learning methods and a probability threshold. In a second step, we successfully used the trained model to predict behaviour in free-ranging cheetahs with the specific intention to detect feeding behaviour and, by adding GPS cluster analysis, identify feeding events. Such feeding events were verified by previously confirmed kill sites in the field (Figure 3).

### 4.1. Data Validation

Observing captive animals inherently goes along with a reduced range of behaviour. In our study, the missing behaviours were mainly hunting, killing, fighting and mating. Nevertheless, our ground-truthed dataset mirrored the natural activity budget of cheetahs, with resting (and observing), walking (moving) and feeding (hunting and eating) making up by far the largest proportion of their daily activities [33,60].

While the six MLAs (LDA, QDA, KNN, CART, SVM, RF) differed in their success in predicting each behaviour, we found the two ensemble learning approaches to be the most promising ones to overall improve precision and recall and a good compromise for the successful identification of the six behaviours. On the other hand, the performance differences of the MLAs allow for a specific use of the best performing algorithm when concentrating on a particular behaviour, as we did in the model application: We found the SVM algorithm to work best in identifying feeding behaviour and used it to determine instances of ‘true feeding’. To account for unverified and mixed behaviour bursts, we implemented a probability threshold of 0.5 [31,61]. This further improved precision and recall.

The results of the LOOCV and the PACV with overall precisions ranging from 57.5% to 91.4% and 69.9% to 92.5%, respectively, are similar to other studies that examined cheetah behaviour predictions: Campbell et al. [30] achieved with the SVM algorithm precision and recall values of >90% for correctly classifying sitting and standing behaviour in tri-axial ACC data of one cheetah. Using the cheetah dataset provided by Campbell et al. [30], Bidder et al. [31] reached precision and recall values of 90% and 97% using the KNN algorithm to predict sitting and standing behaviour. Grünewälder et al. [29] predicted stationary, mobile and feeding behaviour in bi-axial ACC data from six free-ranging cheetahs, for which both training and testing data originated from the same individuals. They achieved overall accuracy between 83.9% and 94.0%, while feeding behaviour had a larger range and was correctly identified in 22.6% to 100.0% of cases [29]. Our precision for feeding behaviour with the uni-axial ACC data using the LOOCV were also high and ranged between 74.4% and 81.6% at a recall between 54.8% and 82.4%. The PACV, i.e., a model trained with n-1 animals and applied to the remaining animal for which the model was not trained, generally leads to a reduction in the prediction accuracy [62,63]. Nevertheless, our precision for the prediction of feeding behaviour were between 65.6% and 91.4% at a recall between 47.0% and 81.1%. Thus, performed in the upper range compared with the results of Grünewälder et al. [29]. 

Behaviours that were not very variable in their movements and lasted relatively long such as resting, walking and trotting (Appendix A Figure A1) reached highest values for precision and recall, whereas more variable and shorter lasting behaviours such as grooming and drinking (Appendix A Figure A1) performed least. The latter two behaviours are highly flexible concerning body postures and duration, thus inter-individual differences are expected to be most pronounced for these behaviours [61]. Feeding behaviour reached intermediate to high precision and recall. Previous studies on other species also had low predictive performance for behaviours that had either structural similarities in movements and/or a high variance within each behaviour (cattle [54], cougar [44], African elephant (*Loxodonta Africana*) [25], roe deer (*Capreolus capreolus*) [28]).

A reduction in the prediction accuracy of inter-individual models such as LOOCV and PACV might also be caused by different sensor tags, variations in individual-specific behaviours, sex or changing environmental conditions [28]. The female cheetah had lower prediction accuracy for walking behaviour and a lower overall prediction accuracy compared to the males. This might be due to having predicted her behaviours by a model trained with only male data or having worn a tag firmware with slightly different ACC setting (10.54 Hz frequency, 3.8 s burst length) from the ones worn by the males (10.0 Hz frequency, 4.0 s burst length). In addition, this female had a higher body mass index and shorter legs than the males (pers. observation), which might have affected gait and/or head movements and amplitudes.

### 4.2. Model Application

We applied our behavioural model to data from free-ranging cheetah. Similar to other studies combining GPS cluster analysis with ACC data [44,45,46], we used the GPS data corresponding to particular behaviour in the ACC data as additional information to support the model predictions. We used the occurrence of GPS clusters to confirm the predicted feeding events because cheetahs that feed on a typical prey animal of 14 to 56 kg [64] are stationary for approximately one to three hours, resulting in GPS clusters [40].

We successfully identified all nine kill sites and 17 of 18 feeding events of the two male coalitions, which we previously verified in the field by using GPS clusters and checking for prey remains. In addition, we detected a feeding event in both members of one coalition that was not confirmed in the field (Figure 2). Perhaps there was a feeding event and kill site in the field, but the prey remains were carried away by other sympatric carnivores such as leopards, brown hyenas (*Parahyaena brunnea*) or black-backed jackals (*Canis mesomelas*).

There might be several reasons for a reduced model performance on free-living conspecifics such as behaviours that do not occur in captivity and thus are not in the training data set or recordings with mixed behaviours within one burst. Nevertheless, we assess our model using the SVM algorithm and the two ensemble learning approaches as sufficiently successful to detect feeding events in free-ranging cheetahs.

The detection of the chasing part of a hunt before a feeding event would increase the reliability of the identified feeding event [44]. Such chasing behaviour, which in our model was represented by the trotting/running behaviour, would be indicated by bursts with high ACC data variation. The chances to identify bursts during chasing behaviour are limited since bursts only cover 3.0% of the time (3.6 s per 120 s), and the time for a chase only lasts on average 37.9 s [65]. Nevertheless, we detected bursts of high ACC variation in 47.1% of the feeding events in the hour before the feeding event started, which represented 77.8% of the kill sites. To avoid missing bursts during chasing time periods, continuous ACC data recordings or the implementation of additional sensors might be useful. For example, Hetem et al. [66] used a combination of body temperature and activity patterns to identify hunting in cheetahs.

Since there are trade-offs between ACC sampling rate, behaviour predictability, memory usage, battery capacity, and data download duration, these parameters need to be considered in the specific context the study is performed in. Setting a low sampling frequency might result in aliasing effects when patterns of higher frequencies occur. Our results on the data validation and the model application using ACC data at approximately 10 Hz sampling frequency at only one axis, however, are particularly promising and important for future applications on free-ranging cheetahs when low sampling frequencies are necessary to ensure long battery and storage lifetime. Similarly, behaviour model accuracies in free-ranging cougars were not affected significantly when decreasing sampling frequency from 65 Hz to 16 Hz, and only a 2% reduction in model accuracy was noted when down-sampling further to 8 Hz [44]. A higher sampling frequency would have disadvantageous practical implications in the field in that the increased data volume would reduce the time period to reach the memory capacity. It would also increase the time needed to download the data from the collars, which is typically done during costly aerial tracking flights. Additionally, our study demonstrated that the use of only one ACC axis was sufficient to reliably identify feeding, resting, trotting/running and walking.

### 4.3. Method Application

Understanding the behaviour of carnivore species is pivotal to their management [67]. Thus, including accelerometry in ecological studies is providing novel insights into carnivore behaviour, with conservation implications. For example, a study on African lions (*Panthera leo*) in a human-dominated landscape in Kenya demonstrated that lions adopted flexible feeding patterns to avoid conflicts with humans, enabling co-existence [68]. Another study showed how human housing density negatively affected the movement and energetics of cougars in the USA, with cougars increasing their kill rate to compensate for anthropogenic disturbance [69]. In the context of human–wildlife conflict, collaring and tracking of conflict species already provided successful conflict mitigation solutions such as adapting livestock management to avoid livestock grazing in high activity areas of carnivores [35] and using geofencing for early warning systems that facilitate quick responses of livestock herders or owners to an approaching carnivore [70]. With human–carnivore conflicts posing one of the greatest challenges in conservation [71], it is essential to develop tools that help improve our understanding of carnivore behaviour and can be applied in conservation to safeguard their future.

## 5. Conclusions

In this study, we demonstrated that MLAs are a useful approach that provide detailed information on the behaviour, particularly the feeding behaviour of cheetahs. We demonstrated the strong potential and practical applicability of ACC data and MLAs for continuous, automated, and high-resolution behaviour monitoring of cheetahs and showed that their feeding behaviour is reliably detectable. Such information can be used to give new insights into the human–wildlife conflict in Namibia and elsewhere in the cheetah range.

## Figures and Tables

**Figure 1 sensors-21-05426-f001:**
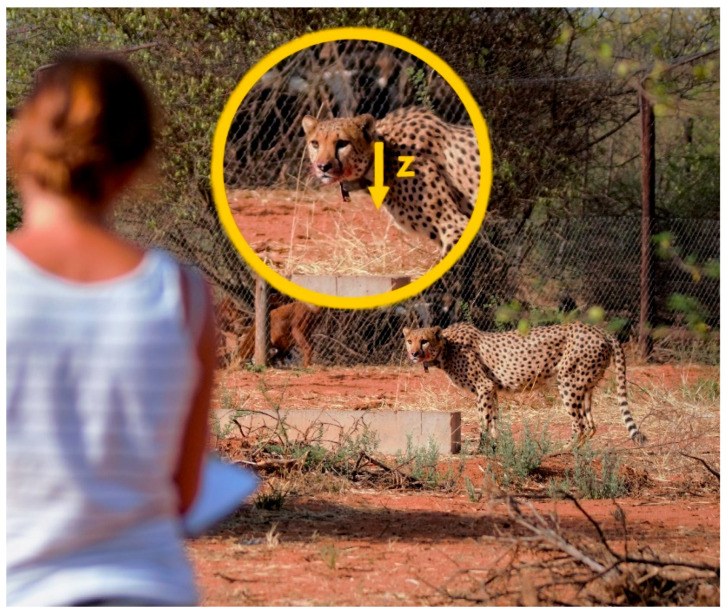
Observer taking notes of a captive cheetah’s behaviour. Captive cheetahs allowed observers to approach to approximately 10–20 m. Collars were set to record acceleration data on the z-axis (yellow arrow in insert), which translates to up-down movements.

**Figure 2 sensors-21-05426-f002:**
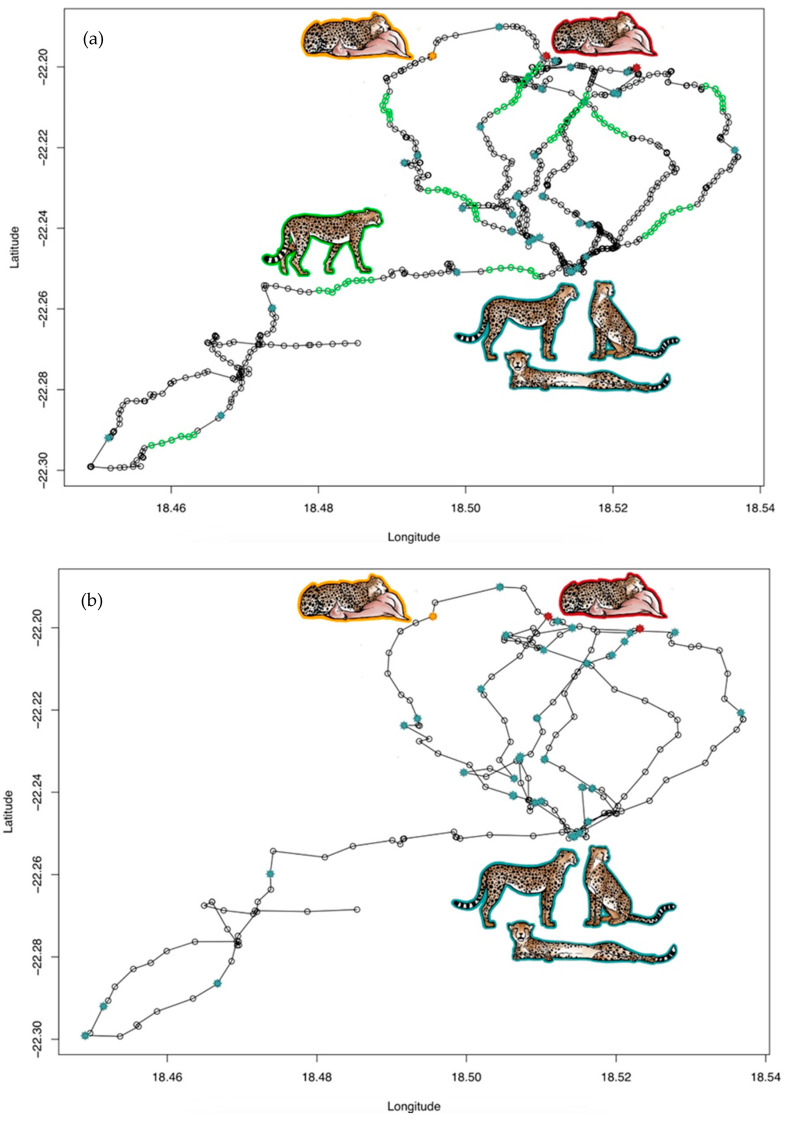
Movement paths from both males of coalition 2 for the same nine days of continuous GPS data with (**a**) collar taking a position every 3 min and (**b**) collar taking a position every 15 min. Blue stars mark the centre of a resting cluster. Green circles mark GPS coordinates representing walking events which could only be calculated for (**a**) since the GPS interval of 15 min in (**b**) was too long to detect these events. Red stars mark the centre of a feeding event that was also confirmed in the field. Orange stars mark the centre of a feeding event that was not confirmed in the field.

**Figure 3 sensors-21-05426-f003:**
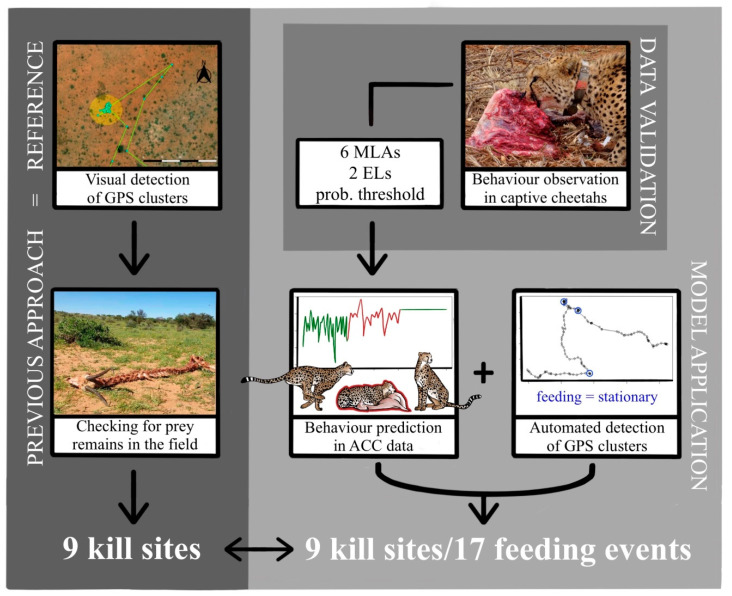
Schematic overview of the methods and results of the study (EL = ensemble learning approach).

**Table 1 sensors-21-05426-t001:** Settings of the accelerometer recordings. The settings for the three captive brothers were programmed at 10.0 Hz and 4.0 s burst length during most of the 36 observation days and at 33.3 Hz and 3.3 s burst length during 7 to 15 days.

Animals	Tag Firmware	Sampling Frequency [Hz]	Burst Length [s]	ACC Samples per Burst & Axis
Captive brothers	A	10.0 or 33.3	4.0 or 3.3	40 or 110
Captive male	A	10	4.0	40
Captive female	B	10.54	3.8	40
Free-ranging males	B	10	3.6	36

**Table 2 sensors-21-05426-t002:** Results of the leave-one-out cross-validations (LOOCV) with a probability threshold of 0.5 for behaviour classification for each of the six machine learning algorithms (MLA), i.e., linear discriminant analysis (LDA), quadratic discriminant analysis (QDA), the k-nearest neighbour (KNN) algorithm, the classification and regression tree (CART) algorithm, support vector machine (SVM) algorithm and the random forest (RF) algorithm, and the two ensemble-learning approaches, i.e., the majority and mean voting. Precisions (PR) across all acceleration data and for the behaviours drinking (D), feeding (F), grooming (G), resting (R), trotting/running (T) and walking (W), and recall (RE) for all behaviours are shown. PR and RE were calculated using the respective confusion matrices, with PR = TP/(TP+FP) and RE = TP/(TP+FN). TP = true positive, FP = false positive, FN = false negative.

MLA	Overall		D		F		G		R		T		W	
	PR	RE	PR	RE	PR	RE	PR	RE	PR	RE	PR	RE	PR	RE
LDA	71.5	55.5	<0.1	0.0	81.6	54.8	71.7	6.8	94.0	87.3	100	89.2	82.0	94.6
QDA	75.5	76.2	41.8	56.2	78.2	71.3	50.0	44.6	99.0	90.7	97.8	99.2	86.1	94.9
KNN	87.5	65.7	12.4	12.4	78.3	74.4	76.4	19.9	94.9	94.8	100	99.2	95.0	93.7
CART	57.5	59.7	<0.1	0.0	74.4	75.4	0.0	0.0	92.1	93.2	98.0	97.7	80.1	91.9
RF	87.4	74.4	82.4	37.8	80.4	77.9	70.8	44.4	97.6	96.8	98.8	98.5	94.1	91.3
SVM	86.9	74.5	79.1	36.8	78.8	82.4	74.0	38.6	96.6	97.6	100	99.2	92.8	92.2
Mean	91.4	67.8	97.1	17.8	81.5	74.9	80.6	25.7	97.9	96.8	99.6	98.1	91.9	93.5
Majority	87.9	70.9	84.4	26.5	79.4	79.4	77.5	29.0	96.5	97.5	99.6	99.2	90.0	94.0

**Table 3 sensors-21-05426-t003:** GPS-informed feeding events based on the 3-min schedule from the field study in 2015, results of the model application for each detected feeding event with the corresponding GPS cluster, information on whether bursts with high acceleration (ACC) variation were detected one hour before the feeding event and whether the two data sets matched in the identified feeding events. Matches are indicated by a check mark in the respective column, a cross indicates no match between the results. A minus indicates that the model did not detect a feeding event and an asterisk * indicates that the model detected a feeding event, but no prey remains were detected at the GPS cluster. The symbol ^#^ indicates instances for which 3-min GPS data were not available for male 3 and the 15-min GPS data of the coalition partner were used.

GPS-Informed Feeding Events					Model Application										Match
	Date	Start	End	Duration [min]	Male	Date	Start	End	Duration [min]	GPS Schedule [min]	Start GPS Cluster	End GPS Cluster	Duration GPS Cluster [min]	High ACC before	
COALITION 1	7 February	19:03	22:06	183	1	7 February	19:28	21:08	100	3	19:03	22:06	183	no	√
					2	7 February	19:30	20:44	73	15	19:15	22:00	165	yes	√
	11 February	05:00	07:30	150	1	11 February	05:02	07:34	152	3	05:03	07:54	171	yes	√
					2	11 February	05:46	07:16	90	15	05:00	07:45	165	no	√
	14 February	19:30	22:30	180	1	14 February	19:40	22:18	158	3	19:27	22:33	186	yes	√
					2	14 February	19:52	22:04	132	15	19:30	22:30	180	no	√
	17 February	18:39	20:03	84	1	17 February	18:30	20:00	90	3	18:30	20:00	90	no	√
					2	17 February	18:45	19:44	59	15	18:45	20:00	75	no	√
	21 March	07:46	09:24	98	1	21 March	07:38	08:58	80	3	07:39	09:24	105	no	√
					2	21 March	00:15	09:10	55	15	08:15	09:15	60	yes	√
COALITION2	13 March	17:06	19:27	141	3	13 March	17:06	19:20	134	3	17:06	19:30	144	no	√
					4	13 March	18:04	18:56	52	15	17:15	19:30	135	yes	√
	16/17 March *	22:48 *	01:25 *	157 *	3	16/17 March	22:54	01:18	144	3	22:48	01:21	153	no	x
					4	17 March	00:50	01:28	38	15	23:00	01:15	135	no	x
	19 March	20:15	00:00	225	3	19 March	20:16	23:20	184	3	20:15	00:06	231	yes	√
					4	19 March	20:26	21:42	76	15	20:15	00:00	225	yes	√
	29/30 April	23:30	01:15	105	3 #	29/30 April	23:20	01:34	134	360	23:30	01:30	120	yes	√
					4	-	-	-	-	15	-	-	-	-	x
	6 May	02:00	05:00	180	3 #	6 May	02:00	05:02	182	360	02:00	05:00	180	no	√
					4	6 May	03:18	04:24	66	15	02:00	05:00	180	no	√

## Data Availability

The data presented in this study are available on request from the corresponding author. The data are not publicly available due to conservation status of the species.

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
