# Peer review of "Using Machine Learning for Remote Behaviour Classification—Verifying Acceleration Data to Infer Feeding Events in Free-Ranging Cheetahs"

_sensors, 2021, doi:10.3390/s21165426_

Round 1

Reviewer 1 Report

This paper detailed investigate the the behaviours in four free-ranging 21 cheetah males using supervised machine learning algorithms. The study has important value for improving our understanding of carnivore behaviour and can be applied in conservation to safeguard their future. Each parts of the manuscript were written in detailed. Some specific recommendations are as follows:

  • The abstract need rewritten for irregular written. The abstract is not a condensed part of the whole article.
  • The article is too long, and many detailed descriptions are not necessary for scientific research papers. Therefore, a lot of content needs to be cut.
  • The conclusion needs to be rewritten, because the core point of the article is not extracted.

Author Response

This paper detailed investigate the the behaviours in four free-ranging cheetah males using supervised machine learning algorithms. The study has important value for improving our understanding of carnivore behaviour and can be applied in conservation to safeguard their future. Each parts of the manuscript were written in detailed. Some specific recommendations are as follows:

  • The abstract need rewritten for irregular written. The abstract is not a condensed part of the whole article.

Reply: We have substantially restructured the methods and results sections of manuscript, focusing more on the model application as suggested by reviewer #3. The abstract now better represents the content of the manuscript and is in line with the focus of it.  

  • The article is too long, and many detailed descriptions are not necessary for scientific research papers. Therefore, a lot of content needs to be cut.

Reply: We shortened substantially the method and result section and have moved some of the content to the Appendix as also suggested by reviewer #3.

  • The conclusion needs to be rewritten, because the core point of the article is not extracted.

Reply: There was no section with a conclusion in the original manuscript. In the revised version we included a conclusion.

Reviewer 2 Report

It is enough to draw reader's interests to detect and identify various behaviors of wild-life species.

The conventional MLAs  have been used to investigate the behaviors of free-ranging cheetahs, and experimental results showed the accuracies of detection and classifications for those methods.

The  followings need to be revised:

In the Section 2 Material and Method, the authors have spent much time to describe the study area, animals, and data collections, analysis. Regardless of the detailed explanation, it is not clear to recognize why the authors have applied 6 MLAs to predict animals behaviors. Further the model testing method need to be represented understandably, i.e.,  the ensemble learning approach and probability threshold in the 2.2.2.

Abstract should be described without any division by (1) Background (2) Methods (3) Results.

In the results, the respresentation of showing the concrete figures 89.6 to 99.6 % needs to be deleted.

The authors  have used much volumn to describe  the section 2 Materials and Methods. For simplicity and clarity of scientific article, it is recommended that the  section 2 is divided the Methods into  conventional MLAs theory and  applications. i.e.., Section 2 and 3..

Author Response

It is enough to draw reader's interests to detect and identify various behaviors of wild-life species.

The conventional MLAs have been used to investigate the behaviors of free-ranging cheetahs, and experimental results showed the accuracies of detection and classifications for those methods.

The followings need to be revised:

In the Section 2 Material and Method, the authors have spent much time to describe the study area, animals, and data collections, analysis. Regardless of the detailed explanation, it is not clear to recognize why the authors have applied 6 MLAs to predict animals behaviors. Further the model testing method need to be represented understandably, i.e., the ensemble learning approach and probability threshold in the 2.2.2.

Reply: We reviewed substantially the material and method section and we declare the application of (different) MLA(s) depending on the certain study task: (copy from 4.1. data validation) “While the six MLAs differed in their success in predicting each behaviour, we found the two ensemble learning approaches to be the most promising ones to overall improve prediction accuracies and a good compromise for the successful identification of all the six behaviours. On the other hand, the performance differences of the MLAs allow for a specific use of the best performing algorithm when concentrating on a particular behaviour, as we did in the model application: In the revised version, we found the SVM algorithm to work best in identifying feeding behaviour and used it to determine instances of ‘true feeding’.

To account for unverified and mixed behaviour bursts we implemented a probability threshold of 0.5 [31,63]. This further improved precision and recall.”

Abstract should be described without any division by (1) Background (2) Methods (3) Results.

Reply: We removed the divisions in the abstract.

In the results, the representation of showing the concrete figures 89.6 to 99.6 % needs to be deleted.

Reply: We deleted these figures from the text.

The authors have used much volumn to describe the section 2 Materials and Methods. For simplicity and clarity of scientific article, it is recommended that the section 2 is divided the Methods into conventional MLAs theory and applications. i.e.., Section 2 and 3.

Reply: We followed this recommendation, which was also suggested by reviewer #3, i.e. we merged two of the three sections to separate the testing and validation part from the application part.

Reviewer 3 Report

The authors develop behaviour recognition algorithms based on accelerometer data collected on captive cheetahs, and then apply these to data from wild cheetahs to detect feeding events. The approach followed is methodical and rigorous, and the focus on feeding event detection has ecological significance.

I have two major comments, one on structure and one on methodology. First, I see two main messages competing for the reader’s attention, which is also reflected in the title: (i) building and validating machine learning algorithms for cheetah behaviour recognition, and (ii) inference of feeding events using accelerometers, and validation of model predictions through GPS data and visual inspection of kill sites. Currently, the vast amount of detail on building and choosing among the machine learning models given in Sections 2.3.1, 2.3.2 and 3.1 (tables 2-4) obfuscate and detract from the part on inference and validation of feeding events. To enhance readability and improve clarity, the authors must choose and give precedence to one of these two messages. My own perspective is that the main message of the paper is about reliable detection of feeding events in cheetahs using accelerometers, and that machine learning is but one constituent of the overall process. Most of the details in the aforementioned sections and tables can be moved to the appendix; Sections 2.3.1 and 2.3.2 can be condensed into a single paragraph with only the most relevant details, and Section 3.1 should only mention the data collected and the ‘winning model’ along with the latter’s performance characterisation while saying that the reader can find more details on model comparison in the appendix. Further, the clarity of the method to infer and then validate a feeding event using GPS data needs to be improved (see comment below for L292-297). Second, some aspects of signal processing and performance characterisation need to be revisited, in particular for the data that was downsampled from 33.3 Hz to 11.1 Hz (see comment for L157-159 below), and application of the probability threshold (see comment for L247-248 below), respectively.

Specific comments are made below:

L46: Technically, accelerometers measure the change in velocity of the sensor; because sensors can be loosely attached, they do not necessarily record the change in velocity of the body part itself, a phenomenon that has received much attention in human studies under the term ‘soft tissue artifact’ or ‘skin motion artifact’ (see, for example, the Discussion section of Dejnabadi, H., Jolles, B. M., & Aminian, K. (2005). A new approach to accurate measurement of uniaxial joint angles based on a combination of accelerometers and gyroscopes. IEEE Transactions on Biomedical Engineering52(8), 1478-1484).

L71: Consider rephrasing ‘animal losses’ as ‘livestock losses’ to be more specific.

L141: Perhaps you meant ‘UTC’ instead of ‘UTM’.

L145: How did you arrive at these particular sampling frequencies to begin with, and was the cut-off frequency of the analog anti-aliasing filter of the tag adapted to remove frequency components of the signal that were higher than half the sampling frequency? For instance, another study of cheetah hunting behaviour used a sampling frequency of 300 Hz (Wilson, A. M., Lowe, J. C., Roskilly, K., Hudson, P. E., Golabek, K. A., & McNutt, J. W. (2013). Locomotion dynamics of hunting in wild cheetahs. Nature498(7453), 185-189). My comment below for L529-530 is also relevant here.

L152-157: I feel that this belongs in the results section.

L157-159: Downsampling is not equivalent to changing sampling frequency, since the downsampled data will still contain (aliased) frequencies from the range 11.1/2 – 33.3/2 Hz. The 33.3 Hz data should first be low-pass filtered with a cut-off of 11.1/2 Hz, and only then downsampled by choosing every third sample.

L165-172: This is an innovative method to annotate behaviours when ACC data is recorded in bursts. What is a ‘sample sheet’ (L169-170), though? Secondly, was it possible using this method to synchronise the precise starting time of behaviour annotation with the starting time of the recording burst? I imagine it must have taken some time (say half or one second) for the observer to detect when the beeping signal started changing its rhythm (L168) – could synchronisation errors (between the beginning of behaviour annotation and start time of a recording burst) resulting from these unavoidable delays (if these were indeed present) have resulted in lower accuracies for shorter-duration behaviours (L472)?

L149-150: What was the rationale for choosing the heave axis instead of surge or sway? Robust postural information is contained in surge acceleration for terrestrial animals wearing collars (two examples: (i) Watanabe, S., Izawa, M., Kato, A., Ropert-Coudert, Y., & Naito, Y. (2005). A new technique for monitoring the detailed behaviour of terrestrial animals: a case study with the domestic cat. Applied Animal Behaviour Science94(1-2), 117-131, and (ii) Section 2.1.1 in Chakravarty, P., Cozzi, G., Ozgul, A., & Aminian, K. (2019). A novel biomechanical approach for animal behaviour recognition using accelerometers. Methods in Ecology and Evolution10(6), 802-814.). I wonder if choosing the surge axis would have improved your drinking classification, despite the reasons mentioned in L472-475.

L210-213: Did you have no cases where the ACC variance threshold was crossed because of scratching or other brief, sudden movements while resting? If you applied an additional threshold during postprocessing (e.g. considering it a possible feeding site only if the number of GPS fixes crossed some minimum number), it would be good to mention it here. Is that what you mean by ‘Once such clusters were identified’ (L212-213)? If yes, it would be good to be more explicit about the identification process. It doesn’t seem like the methodology in L268-273 would address all possible instances of this specific case.

L217-220: Reads as though this should be in the results section.

L234-238: Why were these particular features chosen? If you started from a large set of features and then selected only these six, it would be good to mention how feature selection was done. Finally, please mention the mathematical formulae (or some other appropriate format) used to compute ‘wmz’ and ‘ICVz’ for clarity. I tried to find these details by searching for reference [58] but this is a Masters thesis that I could not find online.

L247-248: Clarify what happens to a burst that is ‘not conclusive’ and ‘dismissed’. Does this mean you remove poorly classified bursts from the dataset? This would artificially inflate model performance. Since you know the behaviour label of this burst through visual observation during groundtruthing, an ‘inconclusive’ burst will have to fall in some non-diagonal entry in the final confusion matrix; it can’t not feature in the confusion matrix at all (what I assume you mean by ‘dismissed’).

L256-257: Reads as though this should be in the results section.

L261: Does ‘prediction accuracy per behaviour’ refer to sensitivity (TP/(TP+FN)) or something else? It would be good to provide its mathematical formula to remove confusion.

L262: I recommend reporting the resulting confusion matrices since (i) different researchers use different classification metrics (e.g. specificity instead of FPR, F-score, etc.) so having the matrices facilitates comparison across studies, and (ii) it becomes easy to see misclassification patterns (e.g. behaviour x often getting misclassified as behaviour y, which helps to understand how to potentially improve the model to prevent this).

L265-266: Not necessarily; the matrix can also be transposed (actual labels along rows and predictions along columns). I suggest removing this.

L281-285: What was the minimum number of closely-spaced fixes required in your automated approach for this set to be called a ‘cluster’? This is related to my comment regarding L210-213.

L292-297: It seems that there are four sets of rules to identify feeding, and it’s unclear which is the reference: (i) the automated approach to define GPS clusters, (ii) physically visiting a site and finding prey remains, (iii) ACC-predicted feeding clusters, and (iv) merger of GPS clusters and ACC-predicted feeding clusters. The current presentation is confusing, and I suggest simplifying and clarifying it, perhaps along the lines of these four categories (if they are indeed true) while clearly specifying the reference that determines whether a site was actually a feeding site or not.

L310-311: Why must raw ACC data be normally distributed, and where do you use this information?

L312: I don’t think ‘exemplary’ is the correct word here. Consider replacing by ‘sample’.

L337: In the caption you mention ‘percentage’ but in the table you report values in fractions. I suggest being consistent by converting values in the table to percentage, which would also improve readability by removing the leading ‘0.’

L363: I suggest converting values in the table to percentage for better readability. Second, I was a bit confused with the entries marked ‘>0.001’. Did you actually mean ‘<0.001’ (i.e., less than 0.1%)? Third, please provide the formula for behaviour-specific PA just like you do for FPR.

L384: I suggest converting values in the table to percentage for better readability and consistency with the text. Please provide the formula for behaviour-specific PA just like you do for FPR.

L470-478: With what other behaviours were drinking, grooming, and feeding typically confounded? Reporting confusion matrices with your results will help disentangle this and provide information on how to improve the classification of these behaviours in future studies.

L489: You don’t ‘verify’ the full behavioural model but only one of the behaviours, i.e., feeding. I would rather align the first sentence with the title of this section, and say something along the lines of, ‘We applied our behavioural model to data from free-ranging cheetah.’

L529-530: Though it’s nice that you comment on sampling frequency choice in your study, one must note that footfall frequency is actually not the highest frequency in the signal. First, several harmonics of the fundamental frequency (5 Hz in your case) are known to exist in the acceleration signal recorded during locomotion (Kavanagh, J. J., & Menz, H. B. (2008). Accelerometry: a technique for quantifying movement patterns during walking. Gait & posture28(1), 1-15). Second, the tag undergoes several mini-impacts against the body of the animal during abrupt movements (e.g. sudden turns, head-jerks, etc.), which generates high frequencies in the acceleration signal (Chakravarty, P., Cozzi, G., Dejnabadi, H., Léziart, P. A., Manser, M., Ozgul, A., & Aminian, K. (2020). Seek and learn: Automated identification of microevents in animal behaviour using envelopes of acceleration data and machine learning. Methods in Ecology and Evolution11(12), 1639-1651). While I am afraid that your signals probably suffer from aliasing due to the low sampling frequency, I also recognise that these concepts require a mathematical/engineering background that ecologists are not trained for. Thus, I propose you present the results as-is while mentioning the caveats of using your sampling frequency, rather than providing this erroneous justification.

L570-571: I suggest deleting the word ‘exemplary’ here (see comment above for L312). Second, it could be useful to support your reasoning in L472-475 by adding more typical postures observed for drinking (a) and grooming (c) in the figure so that the reader can understand how (or whether) the heave axis signal might have made their detection difficult.

Author Response

Comments and Suggestions for Authors

The authors develop behaviour recognition algorithms based on accelerometer data collected on captive cheetahs, and then apply these to data from wild cheetahs to detect feeding events. The approach followed is methodical and rigorous, and the focus on feeding event detection has ecological significance.

I have two major comments, one on structure and one on methodology. First, I see two main messages competing for the reader’s attention, which is also reflected in the title: (i) building and validating machine learning algorithms for cheetah behaviour recognition, and (ii) inference of feeding events using accelerometers, and validation of model predictions through GPS data and visual inspection of kill sites. Currently, the vast amount of detail on building and choosing among the machine learning models given in Sections 2.3.1, 2.3.2 and 3.1 (tables 2-4) obfuscate and detract from the part on inference and validation of feeding events. To enhance readability and improve clarity, the authors must choose and give precedence to one of these two messages. My own perspective is that the main message of the paper is about reliable detection of feeding events in cheetahs using accelerometers, and that machine learning is but one constituent of the overall process. Most of the details in the aforementioned sections and tables can be moved to the appendix; Sections 2.3.1 and 2.3.2 can be condensed into a single paragraph with only the most relevant details, and Section 3.1 should only mention the data collected and the ‘winning model’ along with the latter’s performance characterisation while saying that the reader can find more details on model comparison in the appendix. Further, the clarity of the method to infer and then validate a feeding event using GPS data needs to be improved (see comment below for L292-297).

Reply: Accepting the suggestions of the reviewer, we merged the previous sections 2.3.1 and 2.3.2, shortened and restructured the methods. We rewrote the results and present now only the ‘winning model’ which was used for the application on the free-ranging cheetahs. We moved all other results and tables to the Appendix. The focus is now on the application part of the manuscript.

Second, some aspects of signal processing and performance characterisation need to be revisited, in particular for the data that was downsampled from 33.3 Hz to 11.1 Hz (see comment for L157-159 below), and application of the probability threshold (see comment for L247-248 below), respectively.

Reply: We redid the downsampling of the 33.3 Hz data after using a low-pass filter to remove frequency components of the higher sampling frequency and redid all subsequent analyses. A description for the downsampling was added to the manuscript.

Specific comments are made below:

L46: Technically, accelerometers measure the change in velocity of the sensor; because sensors can be loosely attached, they do not necessarily record the change in velocity of the body part itself, a phenomenon that has received much attention in human studies under the term ‘soft tissue artifact’ or ‘skin motion artifact’ (see, for example, the Discussion section of Dejnabadi, H., Jolles, B. M., & Aminian, K. (2005). A new approach to accurate measurement of uniaxial joint angles based on a combination of accelerometers and gyroscopes. IEEE Transactions on Biomedical Engineering52(8), 1478-1484).

Reply: That’s correct. We modified the sentence accordingly to “ACCs measure the change in velocity of the sensors attached to the body….”

L71: Consider rephrasing ‘animal losses’ as ‘livestock losses’ to be more specific.

Reply: We changed “animal losses” into “such losses” since this sentences refers to the previous ones mentioning “livestock and/or valuable game species” to be predated by cheetahs.

L141: Perhaps you meant ‘UTC’ instead of ‘UTM’.

Reply: Thank you for spotting this typo.

L145: How did you arrive at these particular sampling frequencies to begin with, and was the cut-off frequency of the analog anti-aliasing filter of the tag adapted to remove frequency components of the signal that were higher than half the sampling frequency? For instance, another study of cheetah hunting behaviour used a sampling frequency of 300 Hz (Wilson, A. M., Lowe, J. C., Roskilly, K., Hudson, P. E., Golabek, K. A., & McNutt, J. W. (2013). Locomotion dynamics of hunting in wild cheetahs. Nature498(7453), 185-189). My comment below for L529-530 is also relevant here.

Reply: In wildlife research, there are always big trade-offs between ACC sampling rate, behaviour predictability, memory usage, battery capacity and data download duration. Moreover, the frequency of 10 Hz was a compromise between high frequency (temporal resolution) and long burst length (temporal coverage), since these two parameters could not be adjusted independently in the logger models used: the higher the frequency the shorter the burst length. We did choose 10 Hz based on our experience from previous studies with the same loggers in other species (like fox (Rast et al. 2020 PLoS ONE 15(5): e0227317. https://doi.org/10.1371/journal.pone.0227317), giraffe (Brandes et al. 2021 Sensors, 2229. https://doi.org/10.3390/s21062229), lynx and horse (unbublished)). There, we found that a too short burst length has more devastating effects on behavioural prediction results than too low sampling rate. Therefore, 10 Hz was the possibly lowest sampling rate for the longest possible burst length, focusing that it will be still high enough to distinguish the behaviours we were targeting.

Studies that have directly compared sampling rate and prediction accuracy have been able to show that even lower sampling rates of 5 Hz on sharks (Hounslow et al. 2019 Assessing the effect of sampling frequency on behavioural classification of acceleration data. Journal of Experimental Marine Biology and Ecology, Volume 512, Pages 22-30) or 1 Hz on dingoes (Tatler et al. 2018 High accuracy at low frequency: detailed behavioural classification from accelerometer data. Journal of Experimental Biology, 221, jeb184085. doi:10.1242/jeb.184085rs) are absolutely sufficient for behavioural predictions. In this respect, we are in a safe range with the 10 Hz sampling rate, as the results of this study also prove.

The tags did not have an analog anti-aliasing filter. We do not know if this has an effect on the predictions. We added a sentence to the manuscript to point out the missing filter.

L152-157: I feel that this belongs in the results section.

Reply: As we redid the downsampling of the 33.3 Hz data after using a low-pass filter to remove frequency components of the higher sampling frequency and redid all subsequent analyses, we changed theses lines but moved them also to the result section.

L157-159: Downsampling is not equivalent to changing sampling frequency, since the downsampled data will still contain (aliased) frequencies from the range 11.1/2 – 33.3/2 Hz. The 33.3 Hz data should first be low-pass filtered with a cut-off of 11.1/2 Hz, and only then downsampled by choosing every third sample.

Reply: We redid the downsampling of the 33.3 Hz data after using a low-pass filter to remove frequency components of the higher sampling frequency and redid all subsequent analyses.

L165-172: This is an innovative method to annotate behaviours when ACC data is recorded in bursts. What is a ‘sample sheet’ (L169-170), though? Secondly, was it possible using this method to synchronise the precise starting time of behaviour annotation with the starting time of the recording burst? I imagine it must have taken some time (say half or one second) for the observer to detect when the beeping signal started changing its rhythm (L168) – could synchronisation errors (between the beginning of behaviour annotation and start time of a recording burst) resulting from these unavoidable delays (if these were indeed present) have resulted in lower accuracies for shorter-duration behaviours (L472)?

Reply: We removed “sample” - it was a standard paper sheet.

Since all observations were carried out simultaneously by two observers and the time between the end of one burst and the beginning of the next burst was almost half a minute (enough time to note the behaviour and turn back attention to the animal), we are certain that we were able to observe the complete bursts from start to finish and annotate the behaviour accordingly. We were also very thorough in excluding bursts with mixed behaviours as well as ones we were uncertain about the behaviour. Furthermore, we randomly controlled several bursts per day and animal for such annotation errors. All in all, we cannot rule out this source of error completely, but we believe it to be negligible given the amount of observations.

L149-150: What was the rationale for choosing the heave axis instead of surge or sway? Robust postural information is contained in surge acceleration for terrestrial animals wearing collars (two examples: (i) Watanabe, S., Izawa, M., Kato, A., Ropert-Coudert, Y., & Naito, Y. (2005). A new technique for monitoring the detailed behaviour of terrestrial animals: a case study with the domestic cat. Applied Animal Behaviour Science94(1-2), 117-131, and (ii) Section 2.1.1 in Chakravarty, P., Cozzi, G., Ozgul, A., & Aminian, K. (2019). A novel biomechanical approach for animal behaviour recognition using accelerometers. Methods in Ecology and Evolution10(6), 802-814.). I wonder if choosing the surge axis would have improved your drinking classification, despite the reasons mentioned in L472-475.

Reply: Due to battery lifetime and data download duration constraints, we intended to have only one axis to be active in the collars for later studies on free-ranging cheetahs. We did choose the heave axis, since we were mainly interested in feeding behaviour. Since carnivores typically rip of pieces of meat from their kills which lay on the ground, we assessed the heave axis to be best suitable to display this feeding acceleration. That why we chose the axis most likely to represent this behaviour in a carnivore species.

In a study on dingoes it could be shown that the x-axis (forwards-backwards) is the one with the greatest influence (followed by the heave axis) on the behaviour prediction results, however in this study no feeding behaviour was investigated but several locomotion and resting behaviours (Tatler et al. 2018 High accuracy at low frequency: detailed behavioral classification from accelerometer data. Journal of Experimental Biology, 221, jeb184085. doi: 10.1242 / jeb.184085rs). We were not aware of any other study demonstrating this in a carnivore species, thus we had to rely on this assumption.

Correctly, we could have tested the choice of the axis in preliminary investigations; however, the cheetah`s feeding behaviour (with the jerky biting, tearing and chewing movements of the head) is certainly reflected in all 3 axes, so that we have dispensed with these quite extensive preliminary tests. Even though the heave axis was determined to be the one axis active in the free-ranging cheetahs before this study design was developed, in hindsight it turned out to be suitable to detect feeding behaviour.

Regarding the low prediction accuracy for drinking behaviour, from the application point of view we got lucky to observe drinking at all since cheetahs rarely drink (same goes for the captive ones). If the focus of our study would have been on drinking behaviour we might have chosen another axis or have tested the influence of axis choice in preliminary studies.

L210-213: Did you have no cases where the ACC variance threshold was crossed because of scratching or other brief, sudden movements while resting? If you applied an additional threshold during postprocessing (e.g. considering it a possible feeding site only if the number of GPS fixes crossed some minimum number), it would be good to mention it here. Is that what you mean by ‘Once such clusters were identified’ (L212-213)? If yes, it would be good to be more explicit about the identification process. It doesn’t seem like the methodology in L268-273 would address all possible instances of this specific case.

Reply: On free-ranging cheetahs, we used an "ACC informed GPS schedule" recording (methodology L210-213, see Figure 3 left part). The "ACC informed GPS schedule" is a tool to help the wildlife researchers to maximize the data output by coupling the GPS schedule to the accelerometer measurements: whenever the animal is active above a certain (user-definable) threshold, the tag will intensify (or even start) GPS recordings (GPS high resolution mode).

This dynamic GSP scheduling works with the variance of acceleration, which are calculated onboard the e-obs data loggers. Once a variance threshold is exceeded for a user defined period of time (number of bursts n1) the tag will automatically switch to the higher GPS recording pattern and stay as such until the variance drops again below the set threshold for a given period (number of bursts n2)(n1 and n2 may differ). In our study, the tags were programmed in a way that the ACC variance threshold has to be crossed for three consecutive bursts (means for 6 minutes) to change into the higher GPS schedule. A cheetah scratching during a resting period (and by this exceeding the variance threshold) would still most likely not activate the higher GPS schedule as scratching last only for shorter periods than 6 minutes.

The methodology in L268-273, on the other hand, uses the ACC data to predict behaviour and verify feeding events by matching clusters of feeding behaviour with corresponding GPS clusters. In the process of the GPS cluster analysis, occurring GPS gaps due to the dynamic GPS scheduling were filled up anyways. Nevertheless, we checked for such gaps in the GPS clusters, that were identified to belong to a feeding event and found that they usually did not have gaps >=30 minutes in their GPS data.

With the newly added Figure 3 we tried to give a better overview about our methodology to avoid any confusion.

L217-220: Reads as though this should be in the results section.

Reply: We moved the sentence to the result section.

L234-238: Why were these particular features chosen? If you started from a large set of features and then selected only these six, it would be good to mention how feature selection was done. Finally, please mention the mathematical formulae (or some other appropriate format) used to compute ‘wmz’ and ‘ICVz’ for clarity. I tried to find these details by searching for reference [58] but this is a Masters thesis that I could not find online.

Reply: Since the acceleration was only measured on one axis, the number of possible predictors is much more limited than in many other studies in which 3 axes were used and in which predictors from each single axis, but also those (such as pitch, roll) are used that combine the information from the three axes.

The choice of the six predictors used were based on our experience from studies on other species (Rast et al. 2020, Brandes et al. 2021) but also on other studies in this area (Watanabe et al. 2008 Development of an automatic classification system for eating, ruminating and resting behavior of cattle using an accelerometer. doi: 10.1111 / j.1744-697X.2008.00126.x or Tatler et al. 2018 High accuracy at low frequency: detailed behavioral classification from accelerometer data. Journal of Experimental Biology, 221, jeb184085. Doi: 10.1242 / jeb.184085rs). Five out of six predictors we used can be considered summary statistics because they result in a single number. With the wmz predictor, we tried to result also the Fast Fourier Transformation (of the burst data) in a single number by calculation the weighted mean of the periodogram ordinates for all Fourier frequencies and by this also to include information about rhythmic components of the data. In the revised version, we added the formulas of "wmz" and "ICVz" in the text. We thought about to add a further appendix table with all mathematical formulas, R functions and References for all predictors but refused this option as we only have six predictors and all (except the ‘wmz’) are standard parameters.

L247-248: Clarify what happens to a burst that is ‘not conclusive’ and ‘dismissed’. Does this mean you remove poorly classified bursts from the dataset? This would artificially inflate model performance. Since you know the behaviour label of this burst through visual observation during groundtruthing, an ‘inconclusive’ burst will have to fall in some non-diagonal entry in the final confusion matrix; it can’t not feature in the confusion matrix at all (what I assume you mean by ‘dismissed’).

Reply: The sections mentioning dismissed bursts have been reworded for clarity. These bursts get the label of “not conclusive”. Not conclusive bursts are included in the confusion matrices that you also requested (Table A2-A9 in the revised version) and are also used in the metrics calculation.

L256-257: Reads as though this should be in the results section.

Reply: We removed the sentence to the result section.

L261: Does ‘prediction accuracy per behaviour’ refer to sensitivity (TP/(TP+FN)) or something else? It would be good to provide its mathematical formula to remove confusion.

Reply: Your comments inspired us to discuss the presentation of the performance metrics again. We came to the conclusion that reporting the precision and recall would be more suitable in our context. All sections and tables have been updated accordingly and formulae for both metrics have been added to the manuscript.

L262: I recommend reporting the resulting confusion matrices since (i) different researchers use different classification metrics (e.g. specificity instead of FPR, F-score, etc.) so having the matrices facilitates comparison across studies, and (ii) it becomes easy to see misclassification patterns (e.g. behaviour x often getting misclassified as behaviour y, which helps to understand how to potentially improve the model to prevent this).

Reply: We agree. Confusion matrices have been added to the appendix (Table A2-A9). We decided on only showing those for the probability threshold of 0.5 as show all would be a very long list and only this one was used for carcass detection and should therefore be the most interesting.

L265-266: Not necessarily; the matrix can also be transposed (actual labels along rows and predictions along columns). I suggest removing this.

Reply: We removed the sentence.

L281-285: What was the minimum number of closely-spaced fixes required in your automated approach for this set to be called a ‘cluster’? This is related to my comment regarding L210-213.

Reply: GPS clusters were automatically determined based on the condition that coordinates of consecutive fixes did not exceed a distance of 30 meters to each other and „lasted at least 30 minutes“ (added to the manuscript). Since one coalition member was on a 15-minute GPS schedule, this meant a minimum of three closely-spaced fixes. For the coalition member with the 3-minute schedule this meant 11 closely-spaced fixes.

The methodology in L210-213 describes the checking for prey remains in the field in 2015, where the GPS data were checked visually for GPS clusters.

L292-297: It seems that there are four sets of rules to identify feeding, and it’s unclear which is the reference: (i) the automated approach to define GPS clusters, (ii) physically visiting a site and finding prey remains, (iii) ACC-predicted feeding clusters, and (iv) merger of GPS clusters and ACC-predicted feeding clusters. The current presentation is confusing, and I suggest simplifying and clarifying it, perhaps along the lines of these four categories (if they are indeed true) while clearly specifying the reference that determines whether a site was actually a feeding site or not.

Reply: We tried to describe the method less confusing (L119-124 revised manuscript) hoping that the two approaches to finding feeding events will now become clearer. On the one hand, there was the conventional method: visual check of GPS data for clusters followed by field work to look for prey remains at these places. On the other hand, we have developed an automated method using the ACC data and MLAs to find feeding clusters in the ACC data added by an automated detection of GPS clusters at the same time of the feeding clusters. With the new automated method, we were able to find all the kill sites that we also found with the conventional (previous) method.

For a better overview, we also inserted Figure 3 (a kind of graphic abstract for our manuscript) to make the relation of the two approaches even clearer.

L310-311: Why must raw ACC data be normally distributed, and where do you use this information?

Reply: This step was indeed unnecessary and was deleted.

L312: I don’t think ‘exemplary’ is the correct word here. Consider replacing by ‘sample’.

Reply: We modified the text accordingly.

L337: In the caption you mention ‘percentage’ but in the table you report values in fractions. I suggest being consistent by converting values in the table to percentage, which would also improve readability by removing the leading ‘0.’

Reply: As suggested, we converted all values into percentages.

L363: I suggest converting values in the table to percentage for better readability. Second, I was a bit confused with the entries marked ‘>0.001’. Did you actually mean ‘<0.001’ (i.e., less than 0.1%)? Third, please provide the formula for behaviour-specific PA just like you do for FPR.

Reply: We converted all values into percentages as suggested. Thank you for spotting the wrong symbol ‘>’. Yes, it should be ‘<’ and we changed the values accordingly.

As mentioned before, the metrics have been reworked but formulae were included for the new metrics.

L384: I suggest converting values in the table to percentage for better readability and consistency with the text. Please provide the formula for behaviour-specific PA just like you do for FPR.

Reply: We converted all values into percentages as suggested.

As mentioned before, the metrics have been reworked but formulae were included for the new metrics.

L470-478: With what other behaviours were drinking, grooming, and feeding typically confounded? Reporting confusion matrices with your results will help disentangle this and provide information on how to improve the classification of these behaviours in future studies.

Reply: Drinking seems to be confused with all other behaviours except trotting. We included the most relevant confusion matrices in the appendix so this can be reviewed in more detail.

L489: You don’t ‘verify’ the full behavioural model but only one of the behaviours, i.e., feeding. I would rather align the first sentence with the title of this section, and say something along the lines of, ‘We applied our behavioural model to data from free-ranging cheetah.’

Reply: We modified the sentence as suggested.

L529-530: Though it’s nice that you comment on sampling frequency choice in your study, one must note that footfall frequency is actually not the highest frequency in the signal. First, several harmonics of the fundamental frequency (5 Hz in your case) are known to exist in the acceleration signal recorded during locomotion (Kavanagh, J. J., & Menz, H. B. (2008). Accelerometry: a technique for quantifying movement patterns during walking. Gait & posture, 28(1), 1-15). Second, the tag undergoes several mini-impacts against the body of the animal during abrupt movements (e.g. sudden turns, head-jerks, etc.), which generates high frequencies in the acceleration signal (Chakravarty, P., Cozzi, G., Dejnabadi, H., Léziart, P. A., Manser, M., Ozgul, A., & Aminian, K. (2020). Seek and learn: Automated identification of microevents in animal behaviour using envelopes of acceleration data and machine learning. Methods in Ecology and Evolution, 11(12), 1639-1651). While I am afraid that your signals probably suffer from aliasing due to the low sampling frequency, I also recognise that these concepts require a mathematical/engineering background that ecologists are not trained for. Thus, I propose you present the results as-is while mentioning the caveats of using your sampling frequency, rather than providing this erroneous justification.

Reply: Thank you for your insights. We removed the sentence regarding the Nyquist Sampling Theorem and the footfall frequency. We also added a comment in the discussion mentioning the likely occurrence of the aliasing effect.

L570-571: I suggest deleting the word ‘exemplary’ here (see comment above for L312). Second, it could be useful to support your reasoning in L472-475 by adding more typical postures observed for drinking (a) and grooming (c) in the figure so that the reader can understand how (or whether) the heave axis signal might have made their detection difficult.

Reply: We exchanged “exemplary” with “sample” as above. We also added one more posture for drinking and grooming each in the figure.
